# Continual Lifelong Causal Effect Inference with Real World Evidence

## Abstract

The era of real world evidence has witnessed an increasing availability of observational data, which much facilitates the development of causal effect inference. Although significant advances have been made to overcome the challenges in causal effect estimation, such as missing counterfactual outcomes and selection bias, they only focus on source-specific and stationary observational data. In this paper, we investigate a new research problem of causal effect inference from incrementally available observational data, and present three new evaluation criteria accordingly, including extensibility, adaptability, and accessibility. We propose a Continual Causal Effect Representation Learning method for estimating causal effect with observational data, which are incrementally available from non-stationary data distributions. Instead of having access to all seen observational data, our method only stores a limited subset of feature representations learned from previous data. Combining the selective and balanced representation learning, feature representation distillation, and feature transformation, our method achieves the continual causal effect estimation for new data without compromising the estimation capability for original data. Extensive experiments demonstrate the significance of continual causal effect inference and the effectiveness of our method.

## 1 Introduction

Causal effect inference is a critical research topic across many domains, such as statistics, computer science, public policy, and economics. Randomized controlled trials (RCT) are usually considered as the gold-standard for causal effect inference, which randomly assigns participants into a treatment or control group. As the RCT is conducted, the only expected difference between the treatment and control groups is the outcome variable being studied. However, in reality, randomized controlled trials are always time-consuming and expensive, and thus the study cannot involve many subjects, which may be not representative of the real-world population the intervention would eventually target. Nowadays, estimating causal effects from observational data has become an appealing research direction owing to a large amount of available data and low budget requirements, compared with RCT (Yao et al., 2020). Researchers have developed various strategies for causal effect inference with observational data, such as tree-based methods (Chipman et al., 2010; Wager & Athey, 2018), representation learning methods (Johansson et al., 2016; Li & Fu, 2017; Shalit et al., 2017; Chu et al., 2020), adapting Bayesian algorithms (Alaa & van der Schaar, 2017), generative adversarial nets (Yoon et al., 2018), variational autoencoders (Louizos et al., 2017) and so on.

Although significant advances have been made to overcome the challenges in causal effect estimation with observational data, such as missing counterfactual outcomes and selection bias between treatment and control groups, the existing methods only focus on source-specific and stationary observational data. Such learning strategies assume that all observational data are already available during the training phase and from the only one source. This assumption is unsubstantial in practice due to two reasons. The first one is based on the characteristics of observational data, which are incrementally available from non-stationary data distributions. For instance, the number of electronic medical records in one hospital is growing every day, or the electronic medical records for one disease may be from different hospitals or even different countries. This characteristic implies that one cannot have access to all observational data at one time point and from one single source. The second reason is based on the realistic consideration of accessibility. For example, when the new observational are available, if we want to refine the model previously trained by original data, maybe

the original training data are no longer accessible due to a variety of reasons, e.g., legacy data may be unrecorded, proprietary, too large to store, or subject to privacy constraint (Zhang et al., 2020). This practical concern of accessibility is ubiquitous in various academic and industrial applications. That's what it boiled down to: in the era of big data, we face the new challenges in causal inference with observational data: the **extensibility** for incrementally available observational data, the **adaptability** for extra domain adaptation problem except for the imbalance between treatment and control groups in one source, and the **accessibility** for a huge amount of data.

Existing causal effect inference methods, however, are unable to deal with the aforementioned new challenges, i.e., extensibility, adaptability, and accessibility. Although it is possible to adapt existing causal inference methods to address the new challenges, these adapted methods still have inevitable defects. Three straightforward adaptation strategies are described as follows. (1) If we directly apply the model previously trained based on original data to new observational data, the performance on new task will be very poor due to the domain shift issues among different data sources; (2) If we utilize newly available data to re-train the previously learned model, adapting changes in the data distribution, old knowledge will be completely or partially overwritten by the new one, which can result in severe performance degradation on old tasks. This is the well-known *catastrophic forgetting* problem (McCloskey & Cohen, 1989; French, 1999); (3) To overcome the catastrophic forgetting problem, we may rely on the storage of old data and combine the old and new data together, and then re-train the model from scratch. However, this strategy is memory inefficient and time-consuming, and it brings practical concerns such as copyright or privacy issues when storing data for a long time (Samet et al., 2013). Our empirical evaluations in Section 4 demonstrate that any of these three strategies in combination with the existing causal effect inference methods is deficient.

To address the above issues, we propose a **C**ontinual **C**ausal **E**ffect **R**epresentation **L**earning method (CERL) for estimating causal effect with incrementally available observational data. Instead of having access to all previous observational data, we only store a limited subset of feature representations learned from previous data. Combining the selective and balanced representation learning, feature representation distillation, and feature transformation, our method preserves the knowledge learned from previous data and update the knowledge by leveraging new data, so that it can achieve the continual causal effect estimation for new data without compromising the estimation capability for previous data. To summarize, our main contributions include:

- Our work is the first to introduce the continual lifelong causal effect inference problem for the incrementally available observational data and three corresponding evaluation criteria, i.e., extensibility, adaptability, and accessibility.

- We propose a new framework for continual lifelong causal effect inference based on deep representation learning and continual learning.

- Extensive experiments demonstrate the deficiency of existing methods when facing the incrementally available observational data and our model's outstanding performance.

## 2 BACKGROUND AND PROBLEM STATEMENT

Suppose that the observational data contain $n$ units collected from $d$ different domains and the $d$-th dataset $D_d$ contains the data $\{(x, y, t) | x \in X, y \in Y, t \in T\}$ collected from $d$-th domain, which contains $n_d$ units. Let $X$ denote all observed variables, $Y$ denote the outcomes in the observational data, and $T$ is a binary variable. Let $D_{1:d} = \{D_1, D_2, ..., D_d\}$ be the set of combination of $d$ dataset, separately collected from $d$ different domains. For $d$ datasets $\{D_1, D_2, ..., D_d\}$, they have the common observed variables but due to the fact that they are collected from different domains, they have different distributions with respect to $X$, $Y$, and $T$ in each dataset. Each unit in the observational data received one of two treatments. Let $t_i$ denote the treatment assignment for unit $i$; $i = 1, ..., n$. For binary treatments, $t_i = 1$ is for the treatment group, and $t_i = 0$ for the control group. The outcome for unit $i$ is denoted by $y_t^i$ when treatment $t$ is applied to unit $i$; that is, $y_1^i$ is the potential outcome of unit $i$ in the treatment group and $y_0^i$ is the potential outcome of unit $i$ in the control group. For observational data, only one of the potential outcomes is observed. The observed outcome is called the factual outcome and the remaining unobserved potential outcomes are called counterfactual outcomes.

In this paper, we follow the potential outcome framework for estimating treatment effects (Rubin, 1974; Splawa-Neyman et al., 1990). The individual treatment effect (ITE) for unit $i$ is the difference

between the potential treated and control outcomes, and is defined as $\text{ITE}_i = y_1^i - y_0^i$. The average treatment effect (ATE) is the difference between the mean potential treated and control outcomes, which is defined as $\text{ATE} = \frac{1}{n} \sum_{i=1}^{n} (y_1^i - y_0^i)$.

The success of the potential outcome framework is based on the following assumptions (Imbens & Rubin, 2015), which ensure that the treatment effect can be identified. **Stable Unit Treatment Value Assumption (SUTVA)**: The potential outcomes for any units do not vary with the treatments assigned to other units, and, for each unit, there are no different forms or versions of each treatment level, which lead to different potential outcomes. **Consistency**: The potential outcome of treatment $t$ is equal to the observed outcome if the actual treatment received is $t$. **Positivity**: For any value of $x$, treatment assignment is not deterministic, i.e., $P(T = t | X = x) > 0$, for all $t$ and $x$. **Ignorability**: Given covariates, treatment assignment is independent to the potential outcomes, i.e., $(y_1, y_0) \perp\!\!\!\perp t | x$.

The goal of our work is to develop a novel continual causal inference framework, given new available observational data $D_d$, to estimate the causal effect for newly available data $D_d$ as well as the previous data $D_{1:(d-1)}$ without having access to previous training data in $D_{1:(d-1)}$.

## 3 THE PROPOSED FRAMEWORK

The availability of "real world evidence" is expected to facilitate the development of causal effect inference models for various academic and industrial applications. How to achieve continual learning from incrementally available observational data from non-stationary data domains is a new direction in causal effect inference. Rather than only focusing on handling the selection bias problem, we also need to take into comprehensive consideration three aspects of the model, i.e., the **extensibility** for incrementally available observational data, the **adaptability** for various data sources, and the **accessibility** for a huge amount of data.

We propose the **C**ontinual **C**ausal **E**ffect **R**epresentation **L**earning method (CERL) for estimating causal effect with incrementally available observational data. Based on selective and balanced representation learning for treatment effect estimation, CERL incorporates feature representation distillation to preserve the knowledge learned from previous observational data. Besides, aiming at adapting the updated model to original and new data without having access to the original data, and solving the selection bias between treatment and control groups, we propose one representation transformation function, which maps partial original feature representations into new feature representation space and makes the global feature representation space balanced with respect to treatment and control groups. Therefore, CERL can achieve the continual causal effect estimation for new data and meanwhile preserve the estimation capability for previous data, without the aid of original data.

### 3.1 MODEL ARCHITECTURE

To estimate the incrementally available observational data, the framework of CERL is mainly composed of two components: $(1)$ the baseline causal effect learning model is only for the first available observational data, and thus we don't need to consider the domain shift issue among different data sources. This component is equivalent to the traditional causal effect estimation problem; $(2)$ the continual causal effect learning model is for the sequentially available observational data, where we need to handle more complex issues, such as knowledge transfer, catastrophic forgetting, global representation balance, and memory constraint. We present the details of each component as follows.

#### 3.1.1 THE BASELINE CAUSAL EFFECT LEARNING MODEL

We first describe the baseline causal effect learning model for the initial observational dataset and then bring in subsequent datasets. For causal effect estimation in the initial dataset, it can be transformed into the traditional causal effect estimation problem. Motivated by the empirical success of deep representation learning for counterfactual inference (Shalit et al., 2017; Chu et al., 2020), we propose to learn the selective and balanced feature representations for treated and control units, and then infer the potential outcomes based on learned representation space.

**Learning Selective and Balanced Representation.** Firstly, we adopt a deep feature selection model that enables variable selection in one deep neural network, i.e., $g_{w_1} : X \to R$, where $X$ denotes the original covariate space, $R$ denotes the representation space, and $w_1$ are the learnable parameters in

function $g$. The elastic net regularization term (Zou & Hastie, 2005) is adopted in our model, i.e., $L_{w_1} = \|w_1\|_2^2 + \|w_1\|_1$. Elastic net throughout the fully connected representation layers assigns larger weights to important features. This strategy can effectively filter out the irrelevant variables and highlight the important variables.

Due to the selection bias between treatment and control groups and among the sequential different data sources, the magnitudes of confounders may be significantly different. To effectively eliminate the imbalance caused by the significant difference in magnitudes between treatment and control groups and among different data sources, we propose to use cosine normalization in the last representation layer. In the multi-layer neural networks, we traditionally use dot products between the output vector of the previous layer and the incoming weight vector, and then input the products to the activation function. The result of dot product is unbounded. Cosine normalization uses cosine similarity instead of simple dot products in neural networks, which can bound the pre-activation between $-1$ and $1$. The result could be even smaller when the dimension is high. As a result, the variance can be controlled within a very narrow range (Luo et al., 2018). Cosine normalization is defined as $r = \sigma(r_{norm}) = \sigma\big(\cos(w, x)\big) = \sigma(\frac{w \cdot x}{|w||x|})$, where $r_{norm}$ is the normalized pre-activation, $w$ is the incoming weight vector, $x$ is the input vector, and $\sigma$ is nonlinear activation function.

Motivated by Shalit et al. (2017), we adopt integral probability metrics (IPM) when learning the representation space to balance the treatment and control groups. The IPM measures the divergence between the representation distributions of treatment and control groups, so we want to minimize the IPM to make two distributions more similar. Let $P(g(x)|t = 1)$ and $Q(g(x)|t = 0)$ denote the empirical distributions of the representation vectors for the treatment and control groups, respectively. We adopt the IPM defined in the family of 1-Lipschitz functions, which leads to IPM being the Wasserstein distance (Sriperumbudur et al., 2012; Shalit et al., 2017). In particular, the IPM term with Wasserstein distance is defined as $\text{Wass}(P, Q) = \inf_{k \in \mathcal{K}} \int_{g(x)} \|k(g(x)) - g(x)\| P(g(x)) d(g(x))$, where $\gamma$ denotes the hyper-parameter controlling the trade-off between $\text{Wass}(P, Q)$ and other terms in the final objective function. $\mathcal{K} = \{k | Q(k(g(x))) = P(g(x))\}$ defines the set of push-forward functions that transform the representation distribution of the treatment distribution $P$ to that of the control $Q$ and $g(x) \in \{g(x)_i\}_{i:t_i=1}$.

**Inferring Potential Outcomes.** We aim to learn a function $h_{\theta_1} : R \times T \to Y$ that maps the representation vectors and treatment assignment to the corresponding observed outcomes, and it can be parameterized by deep neural networks. To overcome the risk of losing the influence of $T$ on $R$, $h_{\theta_1}(g_{w_1}(x), t)$ is partitioned into two separate tasks for treatment and control groups, respectively. Each unit is only updated in the task corresponding to its observed treatment. Let $\hat{y}_i = h_{\theta_1}(g_{w_1}(x), t)$ denote the inferred observed outcome of unit $i$ corresponding to factual treatment $t_i$. We minimize the mean squared error in predicting factual outcomes: $L_Y = \frac{1}{n_1} \sum_{i=1}^{n_1} (\hat{y}_i - y_i)^2$.

Putting all the above together, the objective function of our baseline causal effect learning model is: $L = L_Y + \alpha Wass(P, Q) + \lambda L_{w_1}$, where $\alpha$ and $\lambda$ denote the hyper-parameters controlling the trade-off among $Wass(P, Q)$, $L_w$, and $L_Y$ in the objective function.

### 3.1.2 THE SUSTAINABILITY OF MODEL LEARNING

By far, we have built the baseline model for causal effect estimation with observational data from a single source. To avoid catastrophic forgetting when learning new data, we propose to preserve a subset of lower-dimensional feature representations rather than all original covariates. We also can adjust the number of preserved feature representations according to the memory constraint.

After the completion of baseline model training, we store a subset of feature representations $R_1 = \{g_{w_1}(x) | x \in D_1\}$ and the corresponding $\{Y, T\} \in D_1$ as memory $M_1$. The size of stored representation vectors can be reduced to satisfy the pre-specified memory constraint by a herding algorithm (Welling, 2009; Rebuffi et al., 2017). The herding algorithm can create a representative set of samples from distribution and requires fewer samples to achieve a high approximation quality than random subsampling. We run the herding algorithm separately for treatment and control groups to store the same number of feature representations from treatment and control groups. At this point, we only store the memory set $M_1$ and model $g_{w_1}$, without the original data ($D_1$).

### 3.1.3 THE CONTINUAL CAUSAL EFFECT LEARNING MODEL

For now, we have stored memory $M_1$ and baseline model. To continually estimate the causal effect for incrementally available observational data, we incorporate feature representation distillation and feature representation transformation to estimate causal effect for all seen data based on balanced global feature representation space. The framework of CERL is shown in Fig. 1.

**Feature Representation Distillation.** For next available dataset $D_2 = \{(x,y,t)|x \in X, y \in Y, t \in T\}$ collected from second domain, we adopt the same selective representation learning $g_{w_2} : X \rightarrow R_2$ with elastic net regularization ($L_{w_2}$) on new parameters $w_2$. Because we expect our model can estimate causal effect for both previous and new data, we want the new model to inherit some knowledge from previous model. In continual learning, knowledge distillation (Hinton et al., 2015; Li & Hoiem, 2017) is commonly adopted to alleviate the catastrophic forgetting, where knowledge is transferred from one network to another network by encouraging the outputs of the original and new network to be similar. However, for the continual causal effect estimation problem, we focus more on the feature representations, which are required to be balanced between treatment and control, and among different data domains. Inspired by Hou et al. (2019); Dhar et al. (2019); Iscen et al. (2020), we propose feature representation distillation to encourage the representation vector $\{g_{w_1}(x)|x \in D_2\}$ based on baseline model to be similar to the representation vector $\{g_{w_2}(x)|x \in D_2\}$ based on new model by Euclidean distance. This feature distillation can help prevent the learned representations from drifting too much in the new feature representation space. Because we apply the cosine normalization to feature representations and $\|A - B\|^2 = (A - B)^\intercal (A - B) = \|A\|^2 + \|B\|^2 - 2A^\intercal B = 2(1 - cos(A, B))$, the feature representation distillation is defined as $L_{FD}(x) = 1 - cos(g_{w_1}(x), g_{w_2}(x))$, where $x \in D_2$.

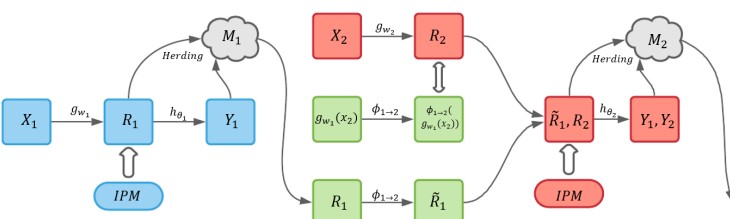

Figure 1: The blue part is baseline causal effect learning model for the first observational data. After baseline model training, store a subset of feature representations $R_1$ into $M_1$ by herding algorithm. The green part helps to map $R_1$ to transformed feature representations $\tilde{R}_1$ compatible with new feature representations space $R_2$. Then the red part is used for continual causal effect estimation based on feature distillation and balanced global feature representation learning for $\tilde{R}_1$ and $R_2$.

**Feature Representation Transformation.** We have previous feature representations $R_1$ stored in $M_1$ and new feature representations $R_2$ extracted from newly available data. $R_1$ and $R_2$ lie in different feature representation space and they are not compatible with each other because they are learned from different models. In addition, we cannot learn the feature representations of previous data from the new model $g_{w_2}$, as we no longer have access to previous data. Therefore, to balance the global feature representation space including previous and new representations between treatment and control groups, a feature transformation function is needed from previous feature representations $R_1$ to transformed feature representations $\tilde{R}_1$ compatible with new feature representations space $R_2$. We define a feature transformation function as $\phi_{1 \rightarrow 2} : R_1 \rightarrow \tilde{R}_1$. We also input the feature representations of new data $D_2$ learned from old model, i.e., $g_{w_1}(x)$, to get the transformed feature representations of new data, i.e., $\phi_{1 \rightarrow 2}(g_{w_1}(x))$. To keep the transformed space compatible with the new feature representation space, we train the transformation function $\phi_{1 \rightarrow 2}$ by making the $\phi_{1 \rightarrow 2}(g_{w_1}(x))$ and $g_{w_2}(x)$ similar, where $x \in D_2$. The loss function is defined as $L_{FT}(x) = 1 - cos(\phi_{1 \rightarrow 2}(g_{w_1}(x)), g_{w_2}(x))$, which is used to train the function $\phi_{1 \rightarrow 2}$ to transform feature representations between different feature spaces. Then, we can attain the transformed old feature representations $\tilde{R}_1 = \phi_{1 \rightarrow 2}(R_1)$, which is in the same space as $R_2$.

**Balancing Global Feature Representation Space.** We have obtained a global feature representation space including the transformed representations of stored old data and new representations of new available data. We adopt the same integral probability metrics as baseline model to make sure that the representation distributions are balanced for treatment and control groups in the global fea-

ture representation space. In addition, we define a potential outcome function $h_{\theta_2} : (\tilde{R}_1, R_2) \times T \to Y$. Let $\hat{y}_i^M = h_{\theta_2}(\phi_{1\to2}(r_i), t)$, where $r_i \in M_1$, and $\hat{y}_j^D = h_{\theta_2}(g_{w_2}(x_j), t)$, where $x_j \in D_2$ denote the inferred observed outcomes. We aim to minimize the mean squared error in predicting factual outcomes for global feature representations including transformed old feature representations and new feature representations: $L_G = \frac{1}{\tilde{n}_1} \sum_{i=1}^{\tilde{n}_1} (\hat{y}_i^M - y_i^M)^2 + \frac{1}{n_2} \sum_{j=1}^{n_2} (\hat{y}_j^D - y_j^D)^2$, where $\tilde{n}_1$ is the number of units stored in $M_1$ by herding algorithm, $y_i^M \in M_1$, and $y_j^D \in D_2$.

In summary, the objective function of our continual causal effect learning model is $L = L_G + \alpha Wass(P, Q) + \lambda L_{w_2} + \beta L_{FD} + \delta L_{FT}$, where $\alpha$, $\lambda$, $\beta$, and $\delta$ denote the hyper-parameters controlling the trade-off among $Wass(P, Q)$, $L_{w_2}$, $L_{FD}$, $L_{FT}$, and $L_G$ in the final objective function.

### 3.2 Overview of CERL

In the above sections, we have provided the baseline and continual causal effect learning models. When the continual causal effect learning model for the second data is trained, we can extract the $R_2 = \{g_{w_2}(x)|x \in D_2\}$ and $\tilde{R}_1 = \{\phi_{1\to2}(r)|r \in M_1\}$. We define a new memory set as $M_2 = \{R_2, Y_2, T_2\} \cup \phi_{1\to2}(M_1)$, where $\phi_{1\to2}(M_1)$ includes $\tilde{R}_1$ and the corresponding $\{Y, T\}$ stored in $M_1$. Similarly, to satisfy the pre-specified memory constraint, $M_2$ can be reduced by conducting the herding algorithm to store the same number of feature representations from treatment and control groups. We only store the new memory set $M_2$ and new model $g_{w_2}$, which are used to train the following model and balance the global feature representation space. It is unnecessary to store the original data ($D_1$ and $D_2$) any longer.

We follow the same procedure for the subsequently available observational data. When we obtain the new observational data $D_d$, we can train $h_{\theta_d}(g_{w_d})$ and $\phi_{d-1\to d} : R_{d-1} \to \tilde{R}_{d-1}$ based on the continual causal effect learning model. Besides, the new memory set is defined as: $M_d = \{R_d, Y_d, T_d\} \cup \phi_{d-1\to d}(M_{d-1})$. So far, our model $h_{\theta_d}(g_{w_d})$ can estimate causal effect for all seen observational data regardless of the data source and it doesn't require access to previous data. The detailed procedures of our CERL method are summarized in Algorithm 1 in Section B of Appendix.

## 4 Experiments

We adapt the traditional benchmarks, i.e., News (Johansson et al., 2016; Schwab et al., 2018) and BlogCatalog (Guo et al., 2020) to continual causal effect estimation. Specifically, we consider three scenarios to represent the different degrees of domain shifts among the incrementally available observational data, including the substantial shift, moderate shift, and no shift. Besides, we generate a series of synthetic datasets and also conduct ablation studies to demonstrate the effectiveness of our model on multiple sequential datasets. The model performance with different numbers of preserved feature representations, and the robustness to hyperparameters are also evaluated.

### 4.1 Dataset Description

We utilize two semi-synthetic benchmarks for the task of continual causal effect estimation, which are based on real-world features, synthesized treatments and outcomes.

**News.** The News dataset consists of 5000 randomly sampled news articles from the NY Times corpus[1]. It simulates the opinions of media consumers on news items. The units are different news items represented by word counts $x_i \in \mathbb{N}^V$ and outcome $y(x_i) \in \mathbb{R}$ is the news item. The intervention $t \in \{0, 1\}$ represents the viewing device, desktop ($t = 0$) or mobile ($t = 1$). We extend the original dataset specification in Johansson et al. (2016); Schwab et al. (2018) to enable the simulation of incrementally available observational data with different degrees of domain shifts. Assuming consumers prefer to read certain media items on specific viewing devices, we train a topic model on a large set of documents and define $z(x)$ as the topic distribution of news item $x$. We define one topic distribution of a randomly sampled document as centroid $z_1^c$ for mobile and the average topic representation of all document as centroid $z_0^c$ for desktop. Therefore, the reader's opinion of news item $x$ on device $t$ is determined by the similarity between $z(x)$ and $z_t^c$, i.e., $y(x_i) = C(z(x)^\intercal z_0^c + t_i \cdot z(x)^\intercal z_1^c) + \epsilon$, where $C = 60$ is a scaling factor and $\epsilon \sim N(0, 1)$.

---

[1] https://archive.ics.uci.edu/ml/datasets/bag+of+words

Table 1: Performance on two sequential data and M=500. We present the mean value of $\sqrt{\epsilon_{\text{PEHE}}}$ and $\epsilon_{\text{ATE}}$ on test sets from two datasets. The standard deviations are tiny. **Lower is better**. **Under no domain shift scenario**, the three strategies and CERL have the similar performance, because the previous and new data are from the same distribution. **Under substantial shift and moderate shift scenarios**, CFR-A performs well on previous data, but significantly declines on new dataset; straxtegy CFR-B shows the catastrophic forgetting problem; CERL has a similar performance to strategy CFR-C, while CERL does not require access to previous data. Besides, the larger domain shift leads to worse performance of CFR-A and CFR-B. CERL has remained stable against shift.

| | | News | | | | | | BlogCatalog | | | | | |
| | | Previous data | | New data | | | | Previous data | | New data | | |
| | Strategy | $\sqrt{\epsilon_{\text{PEHE}}}$ | $\epsilon_{\text{ATE}}$ | $\sqrt{\epsilon_{\text{PEHE}}}$ | $\epsilon_{\text{ATE}}$ | | | $\sqrt{\epsilon_{\text{PEHE}}}$ | $\epsilon_{\text{ATE}}$ | $\sqrt{\epsilon_{\text{PEHE}}}$ | $\epsilon_{\text{ATE}}$ | |
|---|---|---|---|---|---|---|---|---|---|---|---|---|
| **Substantial shift** | CFR-A | 2.49 | 0.80 | **3.62** | **1.18** | ↑ | | 9.92 | 4.25 | **13.65** | **6.21** | ↑ |
| | CFR-B | **3.23** | **1.06** | ↑ | 2.71 | 0.91 | | **14.21** | **6.98** | ↑ | 9.77 | 4.11 |
| | CFR-C | 2.51 | 0.82 | 2.70 | 0.92 | | | 9.93 | 4.24 | 9.77 | 4.12 | |
| | CERL | 2.55 | 0.84 | 2.71 | 0.91 | | | 9.96 | 4.25 | 9.78 | 4.12 | |
| **Moderate shift** | CFR-A | 2.58 | 0.85 | **3.06** | **1.02** | ↑ | | 9.89 | 4.22 | **11.26** | **5.03** | ↑ |
| | CFR-B | **2.98** | **0.99** | ↑ | 2.65 | 0.92 | | **12.35** | **5.67** | ↑ | 9.83 | 4.18 |
| | CFR-C | 2.56 | 0.85 | 2.63 | 0.90 | | | 9.88 | 4.21 | 9.81 | 4.16 | |
| | CERL | 2.59 | 0.86 | 2.66 | 0.92 | | | 9.90 | 4.24 | 9.82 | 4.17 | |
| **No shift** | CFR-A | 2.58 | 0.87 | 2.62 | 0.88 | | | 9.86 | 4.20 | 9.85 | 4.19 | |
| | CFR-B | 2.60 | 0.88 | 2.60 | 0.87 | | | 9.85 | 4.18 | 9.83 | 4.18 | |
| | CFR-C | 2.58 | 0.87 | 2.59 | 0.87 | | | 9.84 | 4.18 | 9.83 | 4.18 | |
| | CERL | 2.59 | 0.87 | 2.60 | 0.87 | | | 9.85 | 4.19 | 9.83 | 4.18 | |

Besides, the intervention $t$ is defined by $p(t = 1|x) = \frac{e^{k \cdot z(x)^{\mathsf{T}} z_1^c}}{e^{k \cdot z(x)^{\mathsf{T}} z_0^c} + e^{k \cdot z(x)^{\mathsf{T}} z_1^c}}$, where $k = 10$ indicates an expected selection bias. In the experiments, 50 LDA topics are learned from the training corpus and 3477 bag-of-words features are in the dataset. To generate two sequential datasets with different domain shifts, we combine the news items belonging to LDA topics from 1 to 25 into first dataset and the news items belonging to LDA topics from 26 to 50 into second dataset. There is no overlap of the LDA topics between the first dataset and second dataset, which is considered as *substantial domain shift*. In addition, the news items belonging to LDA topics from 1 to 35 and items belonging to from 16 to 50 are used to construct the first dataset and second dataset, respectively, which is regarded as *moderate domain shift*. Finally, randomly sampled items from 50 LDA topics compose the first and second dataset, resulting in *no domain shift*, because they are from the same distribution. Under each domain shift scenario and each dataset, we randomly sample 60% and 20% of the units as the training set and validation set and let the remaining be the test set.

**BlogCatalog.** BlogCatalog (Guo et al., 2020) is a blog directory that manages the bloggers and their blogs. In this semi-synthetic dataset, each unit is a blogger and the features are bag-of-words representations of keywords in bloggers' descriptions collected from real-world source. We adopt the same settings and assumptions to simulate the treatment options and outcomes as we do for the News dataset. 50 LDA topics are learned from the training corpus. 5196 units and 2160 bag-of-words features are in the dataset. Similar to the generation procedure of News datasets with domain shifts, we create two datasets for each of the three domain shift scenarios. Under each domain shift scenario and each dataset, we randomly sample 60% and 20% of the units as the training set and validation set and let the remaining be the test set.

## 4.2 Results and Analysis

**Evaluation Metrics.** We adopt two commonly used evaluation metrics. The first one is the error of ATE estimation, which is defined as $\epsilon_{\text{ATE}} = |\text{ATE} - \widehat{\text{ATE}}|$, where ATE is the true value and $\widehat{\text{ATE}}$ is an estimated ATE. The second one is the error of expected precision in estimation of heterogeneous effect (PEHE) Hill (2011), which is defined as $\epsilon_{\text{PEHE}} = \frac{1}{n} \sum_{i=1}^{n} (\text{ITE}_i - \widehat{\text{ITE}}_i)^2$, where $\text{ITE}_i$ is the true ITE for unit $i$ and $\widehat{\text{ITE}}_i$ is an estimated ITE for unit $i$.

We employ three strategies to adapt traditional causal effect estimation models to incrementally available observational data: (A) directly apply the model previously trained based on original data to new observational data; (B) utilize newly available data to fine-tune the previously learned model;

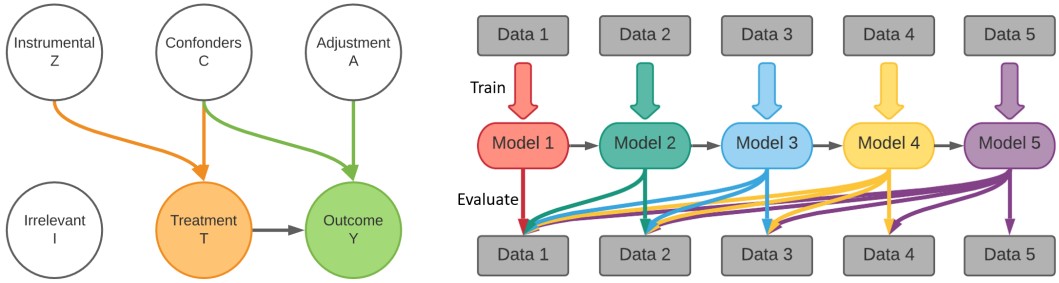

Figure 2: The types of variables.        Figure 3: The work flow of task.

Table 2: Performance on two sequential data and $M = 10000$. We present the mean value of $\sqrt{\epsilon_{\text{PEHE}}}$ and $\epsilon_{\text{ATE}}$ on test sets from two datasets. The standard deviations are tiny. Lower is better.

| Strategy | Previous data | | | New data | | |
|---|---|---|---|---|---|---|
| | $\sqrt{\epsilon_{\text{PEHE}}}$ | $\epsilon_{\text{ATE}}$ | | $\sqrt{\epsilon_{\text{PEHE}}}$ | $\epsilon_{\text{ATE}}$ | |
| CFR-A | 1.47 | 0.35 | | 2.51 | 0.73 | ↑ |
| CFR-B | 1.82 | 0.47 | ↑ | 1.63 | 0.45 | |
| CFR-C | 1.49 | 0.36 | | 1.62 | 0.44 | |
| CERL | 1.49 | 0.37 | | 1.63 | 0.44 | |
| CERL (w/o FRT) | 1.71 | 0.43 | ↑ | 1.63 | 0.44 | |
| CERL (w/o herding) | 1.57 | 0.40 | ↑ | 1.63 | 0.44 | |
| CERL (w/o cosine norm) | 1.51 | 0.38 | ↑ | 1.65 | 0.44 | |

(C) store all previous data and combine with new data to re-train the model from scratch. Among these three strategies, (C) is expected to be the best performer and get the ideal performance with respect to ATE and PEHE, although it needs to take up the most resources (all the data from previous and new dataset). We implement the three strategies based on the counterfactual regression model (CFR) (Shalit et al., 2017), which is a representative causal effect estimation method.

As shown in Table 1, under no domain shift scenario, the three strategies and our model have the similar performance on the News and BlogCatalog datasets, because the previous and new data are from the same distribution. CFR-A, CFR-B, and CERL need less resources than CFR-C. Under substantial shift and moderate shift scenarios, we find strategy CFR-A performs well on previous data, but significantly declines on new dataset; strategy CFR-B shows the catastrophic forgetting problem where the performance on previous dataset is poor; strategy CFR-C performs well on both previous and new data, but it re-trains the whole model using both previous and new data. However, if there is a memory constraint or a barrier to accessing previous data, the strategy CFR-C cannot be conducted. Our CERL has a similar performance to strategy CFR-C, while CERL does not require access to previous data. Besides, by comparing the performance under substantial and moderate shift scenarios, the larger domain shift leads to worse performance of CFR-A and CFR-B. However, no matter what the domain shift is, the performance of our model CERL is consistent with the ideal strategy CFR-C.

### 4.3 MODEL EVALUATION

**Synthetic Dataset.** Our synthetic data include confounders, instrumental, adjustment, and irrelevant variables. The interrelations among these variables, treatments, and outcomes are illustrated in Figure 2. We totally simulate five different data sources with five different multivariate normal distributions to represent the incrementally available observational data. In each data source, we randomly draw 10000 samples including treatment units and control units. Therefore, for five datasets, they have different selection bias, magnitude of covariates, covariance matrices for variables, and number of treatment and control units. To ensure a robust estimation of model performance, for each data source, we repeat the simulation procedure 10 times and obtain 10 synthetic datasets. The details of data simulation are provided in Section A of Appendix.

**Results.** Similar to the experiments for News and BlogCatalog benchmarks, we still utilize two sequential datasets to compare our model with CFR under three strategies on the more complex

synthetic data. As shown in Table 2, the result is consistent with the conclusions on News and Blog-Catalog. Our model's performance demonstrates its superiority over CFR-A and CFR-B. CERL is comparable with CFR-C, while it does not need to have access to the raw data from previous dataset. Besides, we also conduct three ablation studies to test the effectiveness of the important components in CERL, i.e., CERL (w/o FRT), CERL (w/o herding), and CERL (w/o cosine norm). CERL (w/o FRT) is the simplified CERL without the feature representation transformation, which is based on traditional continual learning with knowledge distillation and integral probability metrics. In CERL (w/o FRT), we do not store and transform the previous feature representation into new feature space, and only utilize the knowledge distillation to realize the continual learning task and balance the bias between treatment and control groups with each new data. CERL (w/o herding) adopts random subsampling strategy to select samples into memory, instead of herding algorithm. CERL (w/o cosine norm) removes the cosine normalization in the last representation layer. Table 2 shows that the performance becomes poor after removing anyone in the feature representation transformation, herding, or cosine normalization modules compared to the original CERL. More specifically, after removing the feature representation transformation, $\sqrt{\epsilon_{\text{PEHE}}}$ and $\epsilon_{\text{ATE}}$ increase dramatically, which demonstrates that the knowledge distillation always used in continual learning task is not enough for the continual causal effect estimation. Also, using herding to select a representative set of samples from treatment and control distributions is crucial for the feature representation transformation.

**CERL Performance Evaluation.** As illustrated in Figure 3, the five observational data are incrementally available in sequence, and the model will continue to estimate the causal effect without having access to previous data. We further evaluate the performance of CERL from three perspectives, i.e., the impact of memory constraint, effeteness of cosine normalization, and its robustness to hyper-parameters. As shown in Figure 4 (a) and (b), as the model continually learns a new dataset, every time when finishing training one new dataset, we report the $\sqrt{\epsilon_{\text{PEHE}}}$ and $\epsilon_{\text{ATE}}$ on test sets composed of previous data and new data. Our model with memory constraints has a similar performance to the ideal situation, where all data are available to train the model from scratch. However, our model can effectively save memory space, e.g., when facing the fifth dataset, our model only stores 1000, 5000, or 10000 feature representations, but the ideal situation needs to store $5 \times 10000 = 50000$ observations with all covariates. For the cosine normalization, we perform an ablation study of CERL (M=5000, 5 datasets), where we remove cosine normalization in the representation learning procedure. We find the $\sqrt{\epsilon_{\text{PEHE}}}$ increases from 1.80 and 1.92 and $\epsilon_{\text{ATE}}$ from 0.55 to 0.61. Next, we explore the model's sensitivity to the most important parameter $\alpha$ and $\delta$, which controls the representation balance and representation transformation. From Fig. 4 (c) and (d), we observe that the performance is stable over a large parameter range. In addition, the parameter $\beta$ for feature representation distillation is set to 1 (Rebuffi et al., 2017; Iscen et al., 2020).

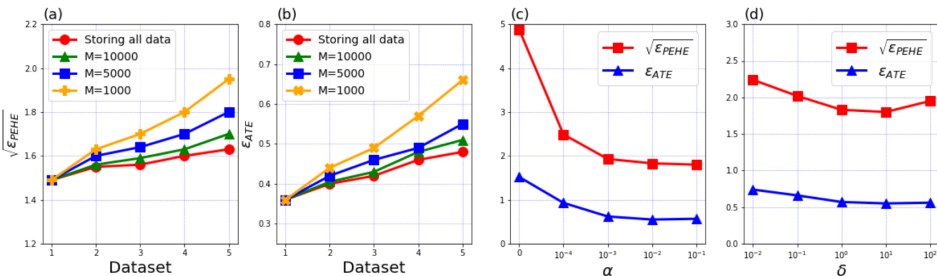

Figure 4: Performance of CERL under different settings.

## 5 CONCLUSION

It is the first time to propose the continual lifelong causal effect inference problem and the corresponding evaluation criteria. As the real world evidence is becoming more prominent, how to integrate and utilize these powerful data for causal effect estimation becomes a new research challenge. To address this challenge, we propose the Continual Causal Effect Representation Learning method for estimating causal effect with observational data, which are incrementally available from non-stationary data distributions. Extensive experiments demonstrate the superiority of our method over baselines for continual causal effect estimation.

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

## A  Simulation Procedure

Our synthetic data include confounders, instrumental, adjustment, and irrelevant variables. The interrelations among these variables, treatments, and outcomes are illustrated in Figure 2. The number of observed variables in the vector $X = (C^\mathsf{T}, Z^\mathsf{T}, I^\mathsf{T}, A^\mathsf{T})^\mathsf{T}$ is set to 100, including 35 confounders in $C$, 35 adjustment variables in $A$, 10 instrumental variables in $Z$, and 20 irrelevant variables in $I$. The model used to generate the continuous outcome variable $Y$ in this simulation is the partially linear regression model, extending the ideas described in Robinson (1988); Jacob et al. (2019); Chu et al. (2020):

$$Y = \tau((C^\mathsf{T}, A^\mathsf{T})^\mathsf{T})T + g((C^\mathsf{T}, A^\mathsf{T})^\mathsf{T}) + \epsilon, \tag{1}$$

where $\epsilon$ are unobserved covariates, which follow a standard normal distribution $N(0,1)$ and $E[\epsilon|C, A, T] = 0$. $T \overset{ind.}{\sim}$ Bernoulli$(e_0((C^\mathsf{T}, Z^\mathsf{T})^\mathsf{T}))$ and $e_0((C^\mathsf{T}, Z^\mathsf{T})^\mathsf{T})$ is the propensity score, which represents the treatment selection bias based on their own confounders $C$ and instrumental variables $Z$. Because we aim to simulate multiple data sources $\{D_d; d = 1, ..., D\}$, the vector of all observed covariates $X = (C^\mathsf{T}, Z^\mathsf{T}, I^\mathsf{T}, A^\mathsf{T})^\mathsf{T}$ is sampled from different multivariate normal distribution with mean vector $\mu_C^d, \mu_Z^d, \mu_I^d$, and $\mu_A^d$ and different random positive definite covariance matrices $\Sigma^d$.

For each data source, except for the different magnitude of mean vector and structure of covariance matrix, the simulation procedure is the same. Let $D$ be the diagonal matrix with the square roots of the diagonal entries of $\Sigma$ on its diagonal, i.e., $D = \sqrt{diag(\sigma)}$, then the correlation matrix is given as:

$$R = D^{-1}\Sigma D^{-1}. \tag{2}$$

We use algorithm 3 in Hardin et al. (2013) to simulate positive definite correlation matrices consisting of different types of variables. Our correlation matrices are based on the hub correlation structure which has a known correlation between a hub variable and each of the remaining variables (Zhang & Horvath, 2005; Langfelder et al., 2008). Each variable in one type of variables is correlated to the hub-variable with decreasing strength from specified maximum correlation to minimum correlation, and different types of variables are generated independently or with weaker correlation among variable types. Defining the first variable as the hub, for the $i$th variable $(i = 2, 3, ..., n)$, the correlation between it and the hub-variable in one type of variables is given as:

$$R_{i,1} = \rho_{\max} - \left(\frac{i-2}{d-2}\right)^\gamma (\rho_{\max} - \rho_{\min}), \tag{3}$$

where $\rho_{\max}$ and $\rho_{\min}$ are specified maximum and minimum correlations, and the rate $\gamma$ controls rate at which correlations decay.

After specifying the relationship between the hub variable and the remaining variables in the same type of variables, we use Toeplitz structure to fill out the remainder of the hub correlation matrix and get the hub-Toeplitz correlation matrix $R_{type}$ for other type of variables. Here, $R$ is the $n \times n$ matrix having the blocks $R_Z, R_C, R_A$, and $R_I$ along the diagonal and zeros at off-diagonal elements. This yields a correlation matrix with nonzero correlations within the same type and zero correlation among other types. The amount of correlations among types which can be added to the positive-definite correlation matrix $R$ is determined by its smallest eigenvalue.

The function $\tau((C^\mathsf{T}, A^\mathsf{T})^\mathsf{T})$ describes the true treatment effect as a function of the values of adjustment variables $A$ and confounders $C$; namely $\tau((C^\mathsf{T}, A^\mathsf{T})^\mathsf{T}) = (\sin((C^\mathsf{T}, A^\mathsf{T})^\mathsf{T} \times b_\tau))^2$ where $b_\tau$ represents weights for every covariate in the function, which is generated by uniform$(0, 1)$. The variable treatment effect implies that its strength differs among the units and is therefore conditioned on $C$ and $A$. The function $g((C^\mathsf{T}, A^\mathsf{T})^\mathsf{T})$ can have an influence on outcome regardless of treatment assignment. It is calculated via a trigonometric function to make the covariates nonlinear, which is defined as $g((C^\mathsf{T}, A^\mathsf{T})^\mathsf{T}) = (\cos((C^\mathsf{T}, A^\mathsf{T})^\mathsf{T} \times b_g))^2$. Here, $b_g$ represents a weight for each covariate in this function, which is generated by uniform$(0, 1)$. The bias is attributed to unobserved covariates which follow a random normal distribution $N(0, 1)$. The treatment assignment $T$ follows the Bernoulli distribution, i.e., $T \overset{ind.}{\sim}$ Bernoulli$(e_0((C^\mathsf{T}, Z^\mathsf{T})^\mathsf{T}))$ with probability

$e_0((C^\intercal, Z^\intercal)^\intercal) = \Phi(\frac{a-\mu(a)}{\sigma(a)})$, where $e_0((C^\intercal, Z^\intercal)^\intercal)$ represents the propensity score, which is the cumulative distribution function for a standard normal random variable based on confounders $C$ and instrumental variables $Z$, i.e., $a = \sin((C^\intercal, Z^\intercal)^\intercal \times b_a)$, where $b_a$ is generated by uniform$(0, 1)$.

We totally simulate five different data sources with five different multivariate normal distributions to represent the incrementally available observational data. In each data source, we randomly draw 10000 samples including treatment units and control units. Therefore, for five datasets, they have different selection bias, magnitude of covariates, covariance matrices for variables, and number of treatment and control units. To ensure a robust estimation of model performance, for each data source, we repeat the simulation procedure 10 times and obtain 10 synthetic datasets.

## B    ALGORITHM 1

---

**Algorithm 1** Continual Causal Effect Representation Learning

---

**Data:** Given $d$ incrementally available observational data from $D_1$ to $D_d$

**if** $\{x, y, t\} \in D_1$ **then**

  *** Train baseline causal effect model $h_{\theta_1}(g_{w_1})$ ***

  $w_1, \theta_1 = \text{OPTIMIZE}(L_Y + \alpha Wass(P, Q) + \lambda L_{w_1})$

  $R_1 = \{g_{w_1}(x) | x \in D_1\}$

  $M_1 = \text{HERDING}\{R_1, Y_1, T_1\}$

**else**

  **for** $\{x, y, t\} \in D_2, ..., D_d$ **do**

    *** Train continual causal effect model $h_{\theta_d}(g_{w_d})$ ***

    $w_d, \theta_d, \phi_{d-1 \to d} = \text{OPTIMIZE}(L_G + \alpha Wass(P, Q) + \lambda L_{w_2} + \beta L_{FD} + \delta L_{FT})$

    $\tilde{R}_{d-1} = \phi_{d-1 \to d}(R_{d-1})$

    $R_d = \{g_{w_d}(x) | x \in D_d\}$

    $M_d = \text{HERDING}(\{R_d, Y_d, T_d\} \cup \{\tilde{R}_{d-1}, Y_{d-1} \in M_{d-1}, T_{d-1} \in M_{d-1}\})$

  **end**

**end**

---

