# OpenReview forum: "Continual Lifelong Causal Effect Inference with Real World Evidence"
_ICLR.cc/2021/Conference — Reject_

### Official Review · AnonReviewer4 · 2020-10-19
**Important problem with some good ideas but the current paper needs substantial revision.**

**Rating:** 2
**Confidence:** 4

**Review:**

The authors propose to estimate individualized treatment effects in a setting they describe as continual learning, batches of data becoming available over time. The technical proposal is to store a feature representation of the presently available data to be updated with new data in the future -- a technique that, together with other regularization methods for selection bias and variable selection, is shown to perform well on synthetic experiments.

Causal inference is increasingly relevant as machine learning assists decision-making, and an important aspect of this process is to handle information streams over time and update accordingly, just as humans would be expected to. In the context of causality though, I believe this requires a more detailed motivation. Causal effects by their very definition exhibit invariance to interventions on observed variables, and if properly estimated, these should not vary between well-behaved environments. With enough data a priori there should not be large changes in estimation in new environments. This may certainly change if the underlying causal mechanisms change and is a problem that has been studied as data fusion, yet this line of research is not mentioned in the paper.

Writing needs to improve substantially. The use of the words extensibility, adaptability, and accessibility is grammatically incorrect, and the context in which they are used does not clarify the meaning of these ideas either. As I understand it, adaptability refers to domain adaptation yet how domains differ, how to merge them depending on their differences, or even an investigation of these problems is not discussed. Accessibility refers to the amounts of data one needs to deal with. This does not strike me as a problem in causal inference where datasets are typically small. Perhaps examples of applications where each of these aspects is a concern would be helpful.

More importantly, the notation is counter-intuitive and inconsistent. For instance, \mathcal X is defined as the set of all observed variables, but then a specific realization x is written to be an element of \mathcal X (as if \mathcal X was a measurable space of possible realizations of a random variable). Both interpretations cannot be correct. The notation \mathcal X_d is similarly ambiguous, does this mean each domain has different sets of variables or that the spaces in which they are defined differ?

The experiments are underwhelming and certainly do not back up the claims made in the introduction. Most benchmarks for individualized treatment effects are semi-synthetic (it would be advisable to stick to these data generating processes while modifying them to highlight specific features of your problem). Many more competing algorithms could be evaluated.

---

> ### Author Response · Authors · 2020-11-25
> **Response to Reviewer 4**
>
> We very much appreciate your useful suggestions and we have addressed these points in the revised version.
>
> **Motivation:**
>
> In our work, we want to emphasize the model's ability of continual causal effect estimation with increasingly available observational data, not only the data fusion of non-stationary datasets. The continual causal effect estimation is suitable for many real-world applications that continuously collect data from multiple sources. For instance, the healthcare data are usually composed of multiple databases collected from different hospitals or other organizations even across multiple countries. And the data are incessantly growing to update the databases in real time.
>
> **Extensibility, Adaptability, and Accessibility:**
>
>
> Extensibility means that the observational data are incrementally available as time goes by. The data cannot be obtained at one time.
>
> Adaptability means that these incrementally available datasets maybe come from different domains. We need to address the domain shift problem in addition to the imbalance between treatment and control groups in every single domain. In particular, we utilize the Feature Representation Distillation to solve the domain shift issue. The feature representations stored in memory and new feature representations extracted from newly available data lie in different feature representation spaces and they are not compatible with each other, because they are learned from different models. The feature transformation function transforms previous feature representations to a new space that is compatible with new feature representations space. In the revised manuscript, we have added new experiments to simulate different degrees of domain shift, i.e., substantial shift, moderate shift, and no shift.
>
> Accessibility: Although the existing benchmarks used in the paper are typically small, the proposed method could be potentially applied to large-scale real-world data such as electronic health records.

---

> > ### Author Response · Authors · 2020-11-25
> > **Response to Reviewer 4 - Continued**
> >
> > **Experiments:**
> >
> > As for the issue about limited experimental evidence, we have added two semi-synthetic benchmarks, i.e., News {johansson2016learning, schwab2018perfect} and BlogCatalog  {guo2020learning} for continual causal effect estimation. For both datasets, we consider three scenarios to represent the different degrees of domain shifts among the incrementally available observational data, i.e., substantial shift, moderate shift, and no shift.
> > Take the News dataset as one example. To generate two sequential datasets with different domain shifts, we combine the news items belonging to LDA topics from $1$ to $25$ into the first dataset and the news items belonging to LDA topics from $26$ to $50$ into the second dataset. There is no overlap of the LDA topics between the first and second datasets, which is considered as substantial shift. In addition, the news items belonging to LDA topics from $1$ to $35$ and items belonging to topics $16$ to $50$ are used to construct the first and second datasets, respectively, which create the moderate shift scenario. Finally, randomly sampled items from $50$ LDA topics compose the first and second dataset with no shift, because they are from the same distribution. Under each domain shift scenario and each dataset, we randomly sample $60\%$ and $20\%$ of the units as the training set and validation set and let the remaining be the test set.
> >
> > |||   News      ||||||BlogCatalog       ||||||
> > |: - :|: - :|: - :|: - :|: - :|: - :|: - :|: - :|: - :|: - :|: - :|: - :|: - :|: - :|
> > |||   Previous data|||New data|||Previous  data|||New data|||
> > ||Strategy|  $\sqrt{\epsilon_\text{PEHE}}$|$\epsilon_\text{ATE}$| |$\sqrt{\epsilon_\text{PEHE}}$|$\epsilon_\text{ATE}$| |$\sqrt{\epsilon_\text{PEHE}}$|$\epsilon_\text{ATE}$| |$\sqrt{\epsilon_\text{PEHE}}$|$\epsilon_\text{ATE}$| |
> > |Substantial | CFR-A |2.49 | 0.80 |    | $\textbf{3.62}$ | $\textbf{1.18}$ | $ \uparrow$ | 9.92 | 4.25 |   | $\textbf{13.65}$ | $\textbf{6.21}$ |  $ \uparrow$|
> > |  shift | CFR-B  |$\textbf{3.23}$ | $\textbf{1.06}$| $ \uparrow$  | 2.71 | 0.91 |   |  $\textbf{14.21}$ | $\textbf{6.98}$ | $ \uparrow$ | 9.77 | 4.11 |
> > |     | CFR-C  | 2.51 | 0.82 |    | 2.70 | 0.92 |    |  9.93 | 4.24|   | 9.77 | 4.12 |
> > |   | CERL  |2.55 | 0.84 |    | 2.71 | 0.91 |   | 9.96 | 4.25 |    | 9.78 | 4.12 |
> > |Moderate | CFR-A  |2.58 |0.85  |    | $\textbf{3.06}$ | $\textbf{1.02}$ | $ \uparrow$ | 9.89 | 4.22 |   | $\textbf{11.26}$ |$ \textbf{5.03}$| $ \uparrow$ |
> > | shift | CFR-B  |$\textbf{2.98}$ |$ \textbf{0.99}$ | $ \uparrow$  | 2.65| 0.92 |   | $\textbf{12.35}$ | $\textbf{5.67} $| $ \uparrow$ | 9.83 | 4.18 |
> >  | | CFR-C  | 2.56 | 0.85 |    | 2.63 | 0.90 |    | 9.88 | 4.21|   | 9.81 | 4.16 |
> >  | | CERL  |2.59 | 0.86 |    | 2.66 | 0.92 |   | 9.90 | 4.24 |    | 9.82 | 4.17 |
> > | No  | CFR-A | 2.58 | 0.87 |   | 2.62 | 0.88 |    | 9.86 | 4.20 |   | 9.85 | 4.19 |
> > |shift  | CFR-B | 2.60 | 0.88 |   | 2.60 | 0.87 |   | 9.85 | 4.18 |   | 9.83 | 4.18 |
> > | | CFR-C | 2.58 | 0.87 |    | 2.59 | 0.87 |    | 9.84 | 4.18|   | 9.83 | 4.18 |
> > | | CERL | 2.59 | 0.87 |    | 2.60 | 0.87 |   | 9.85 | 4.19|   | 9.83 | 4.18 |
> >
> > The Table shows that, under no domain shift scenario, the three strategies and our model have a similar performance on the News and BlogCatalog datasets, because the previous and new data are from the same distribution. CFR-A, CFR-B, and CERL need fewer resources than CFR-C. Under substantial shift and moderate shift scenarios, we find strategy CFR-A performs well on previous data, but significantly declines on new dataset; strategy CFR-B shows the catastrophic forgetting problem where the performance on the previous dataset is poor; strategy CFR-C performs well on both previous and new data, but it re-trains the whole model using both the previous and new data. However, if there is a memory constraint or a barrier to accessing previous data, the strategy CFR-C cannot be conducted. Our CERL has a similar performance to strategy CFR-C, while CERL does not require access to previous data. Besides, by comparing the performance under substantial and moderate shift scenarios, the larger domain shift leads to worse performance of CFR-A and CFR-B. However, no matter what the domain shift is, the performance of our model CERL is consistent with the ideal strategy CFR-C.

---

> > > ### Author Response · Authors · 2020-11-25
> > > **Response to Reviewer 4 - Continued**
> > >
> > > **Experiments:**
> > >
> > > We have conducted three ablation studies to evaluate the effectiveness of the important components in CERL, i.e., CERL (w/o FRT), CERL (w/o herding), and CERL (w/o cosine norm). CERL (w/o FRT) is the simplified CERL without the feature representation transformation, which is based on traditional continual learning with knowledge distillation and integral probability metrics. In CERL (w/o FRT), we do not store and transform the previous feature representation into new feature space, and only utilize the knowledge distillation to realize the continual learning task and balance the bias between treatment and control with each new data. CERL (w/o herding) adopts random subsampling strategy to select samples into memory, instead of herding algorithm. CERL (w/o cosine norm) removes the cosine normalization in the last representation layer. From the Table, the performance becomes poor after removing anyone in the feature representation transformation, herding, or cosine normalization modules compared to the original CERL. More specifically, after removing the feature representation transformation, $\sqrt{\epsilon_\text{PEHE}}$ and $\epsilon_\text{ATE}$ increase dramatically, which demonstrates that the knowledge distillation always used in continual learning task is not enough for the continual causal effect estimation. In addition, using herding to select a representative set of samples from treatment and control distributions is crucial for the feature representation transformation.
> > >
> > > ||Previous data|||New data|||
> > > |: ----- :|: ----- :|: ----- :|: ----- :|: ----- :|: ----- :|: ----- :|
> > > | Strategy     | $\sqrt{\epsilon_\text{PEHE}}$   | $\epsilon_\text{ATE}$ |   | $\sqrt{\epsilon_\text{PEHE}}$   | $\epsilon_\text{ATE}$ |
> > > |    CFR-A | 1.47 | 0.35 |   |  $\textbf{2.51}$ |  $\textbf{0.73}$ | $ \uparrow$ |
> > >  |   CFR-B |  $\textbf{1.82}$ |  $\textbf{0.47}$ | $ \uparrow$ | 1.63 | 0.45 |
> > >  |   CFR-C | 1.49 | 0.36 |   | 1.62 | 0.44 |
> > > |   CERL | 1.49 | 0.37 |    | 1.63 | 0.44 |
> > >  |    CERL (w/o FRT) |  $\textbf{1.71}$ |  $\textbf{0.43}$ |$ \uparrow$  | 1.63 | 0.44|
> > >   |   CERL (w/o herding) |  $\textbf{1.57}$ |  $\textbf{0.40}$ | $ \uparrow$ | 1.63| 0.44|
> > >    |  CERL (w/o cosine norm) |  $\textbf{1.51}$ |  $\textbf{0.38}$ | $ \uparrow$ | 1.65 | 0.44|
> > >
> > > **Notation**:
> > >
> > > We fully agree with you that the notations in the previous version are improper. Sorry for the confusion about the notation $\mathcal{X}_d$. We have deleted the notation $\mathcal{X}_d$. The observational data contain $n$ units collected from $d$ different domains and the $d$-th dataset $D_d$ contains the data $\\{(x,y,t) |x\in X, y\in Y, t\in T \\}$ collected from $d$-th domain. For $d$ datasets $\\{D_1, D_2,..., D_d \\}$, they have the common observed variables but due to the fact that they are collected from different domains, they have different distributions with respect to $X$, $Y$, and $T$ in each dataset. For the whole data, the feature space is consistent. $\\{D_1, D_2,..., D_d \\}$ are only the subsets of the whole population.

---

### Official Review · AnonReviewer1 · 2020-10-26

**Rating:** 3
**Confidence:** 4

**Review:**


This paper considers adopting continual learning on the problem of causal effect estimation. The paper combines methods and algorithms for storing feature representation and representative samples (herding algorithm), avoiding drifting feature representation when new data is learned (feature representation distillation), balanced representation by regularization, etc. Consequently, the paper presents a system that makes use of existing methods as a loss function (the sum of losses and regularization terms).

It is difficult to observe the novelty of the method---there is no apparent challenge arose in combining these methods as they can just be simply added. Further, the development of this framework for causal inference is mostly orthogonal to the problem of causal effect estimation (and the use of regularization does not add any novelty.) In addition to the lack of novelty, the paper has many non-negligible mistakes in describing causal inference. For example, consider a phrase "selection bias" between "treatment and control groups" in the first sentence of the second paragraph. "Treatment and control groups" usually are used to describe two groups under the context of RCT. Further, the bias we are talking in the paper is a "confounding bias" not a "selection bias". In Section 2, "Each unit ... received ... one of ... treatments" is also relevant to RCT not observational data. There are many other places the authors mentioning "selection bias".

Question:
What is "Real world evidence"?, which is used in the title, abstract, conclusion, and at the beginning of section 3? Is it just observational data?

---

> ### Author Response · Authors · 2020-11-25
> **Response to Reviewer 3**
>
> Thank you for your comments. We respond to your individual concerns below.
>
> **Novelty:**
>
> Our work is the first to introduce the continual causal effect estimation problem for the continuously collected observational data. As discussed in Section 1, this new problem setting involves several unprecedented research challenges, such as mitigating domain shifts in the incrementally available observational data.
>
> **" Non-negligible Mistakes":**
>
> Comment: "Treatment and control groups" usually are used to describe two groups under the context of RCT.
>
> Response: In observational studies with binary treatments, treatment group and control group are commonly used terminology in literature [Ref-1, Ref-2, Ref-3, Ref-4, Ref-5].
>
> Comment: Further, the bias we are talking about in the paper is a "confounding bias" not a "selection bias". In Section 2, "Each unit ... received ... one of ... treatments" is also relevant to RCT, not observational data.
>
> Response: In the field of causal inference, selection bias in observational data has been investigated for over three decades [Ref-4, Ref-5]. Confounder variables affect units’ treatment choices, which leads to the selection bias. "Each unit ... received ... one of ... treatments" is a standard-setting in observational studies with binary treatments.
>
> [Ref-1] Alaa, Ahmed M., and Mihaela van der Schaar. "Bayesian inference of individualized treatment effects using multi-task gaussian processes." Advances in Neural Information Processing Systems. 2017.
>
> [Ref-2] Shalit, Uri, Fredrik D. Johansson, and David Sontag. "Estimating individual treatment effect: generalization bounds and algorithms." International Conference on Machine Learning, 2017.
>
> [Ref-3] Louizos, Christos, et al. "Causal effect inference with deep latent-variable models." Advances in Neural Information Processing Systems. 2017.
>
> [Ref-4] Yao, Liuyi, et al. "A Survey on Causal Inference." arXiv preprint arXiv:2002.02770 (2020).
>
> [Ref-5] Guo, Ruocheng, et al. "A survey of learning causality with data: Problems and methods." ACM Computing Surveys (CSUR) 53.4 (2020): 1-37.
>
> **What is "Real world evidence"?**
>
>  Real world evidence (RWE) means evidence obtained from real world data (RWD), which are observational data obtained outside the context of randomized controlled trials (RCTs) and generated during routine clinical practice.

---

### Official Review · AnonReviewer2 · 2020-10-28
**reasonable method; limited experimental evidence;**

**Rating:** 4
**Confidence:** 4

**Review:**

I think the problem proposed in the paper is interesting and the method is moderately novel. I did not find the paper's writing very helpful. The method seems reasonably motivated but the discussion around it does not provide enough intuition about the influence of the different pieces on the final prediction, and no ablation experiment are provided.

My main issue with the paper is the limited experimental evidence.

1. Without any real data  (at least semi-synthetic) experiments, I do not think there is a good way to evaluate the significance of the problem and the proposed method.

2. The single synthetic experiment in the paper is  not sufficient to evaluate the effectiveness of the method. I believe at a minimum, different synthetic experiments with different structural models, dataset shifts and rate of shift over time should be considered.

3. No ablation studies are done and no real baseline is considered. For example, a non-trivial baseline would be to do use a traditional continual learning objective with distillation with a new treatment-control population balancing objective with each new dataset. The value of the proposed method's representation-herding would then become clear.


If the authors address these issues, I'm happy to change my score.

---

> ### Author Response · Authors · 2020-11-25
> **Response to Reviewer 2**
>
> **Review 1,2:**
>
> As for the issue about limited experimental evidence, we have added two semi-synthetic benchmarks, i.e., News {johansson2016learning, schwab2018perfect} and BlogCatalog  {guo2020learning} for continual causal effect estimation. For both datasets, we consider three scenarios to represent the different degrees of domain shifts among the incrementally available observational data, i.e., substantial shift, moderate shift, and no shift.
> Take the News dataset as one example. To generate two sequential datasets with different domain shifts, we combine the news items belonging to LDA topics from $1$ to $25$ into the first dataset and the news items belonging to LDA topics from $26$ to $50$ into the second dataset. There is no overlap of the LDA topics between the first and second datasets, which is considered as substantial shift. In addition, the news items belonging to LDA topics from $1$ to $35$ and items belonging to topics $16$ to $50$ are used to construct the first and second datasets, respectively, which create the moderate shift scenario. Finally, randomly sampled items from $50$ LDA topics compose the first and second dataset with no shift, because they are from the same distribution. Under each domain shift scenario and each dataset, we randomly sample $60\%$ and $20\%$ of the units as the training set and validation set and let the remaining be the test set.
>
> |||   News      ||||||BlogCatalog       ||||||
> |: - :|: - :|: - :|: - :|: - :|: - :|: - :|: - :|: - :|: - :|: - :|: - :|: - :|: - :|
> |||   Previous data|||New data|||Previous  data|||New data|||
> ||Strategy|  $\sqrt{\epsilon_\text{PEHE}}$|$\epsilon_\text{ATE}$| |$\sqrt{\epsilon_\text{PEHE}}$|$\epsilon_\text{ATE}$| |$\sqrt{\epsilon_\text{PEHE}}$|$\epsilon_\text{ATE}$| |$\sqrt{\epsilon_\text{PEHE}}$|$\epsilon_\text{ATE}$| |
> |Substantial | CFR-A |2.49 | 0.80 |    | $\textbf{3.62}$ | $\textbf{1.18}$ | $ \uparrow$ | 9.92 | 4.25 |   | $\textbf{13.65}$ | $\textbf{6.21}$ |  $ \uparrow$|
> |  shift | CFR-B  |$\textbf{3.23}$ | $\textbf{1.06}$| $ \uparrow$  | 2.71 | 0.91 |   |  $\textbf{14.21}$ | $\textbf{6.98}$ | $ \uparrow$ | 9.77 | 4.11 |
> |     | CFR-C  | 2.51 | 0.82 |    | 2.70 | 0.92 |    |  9.93 | 4.24|   | 9.77 | 4.12 |
> |   | CERL  |2.55 | 0.84 |    | 2.71 | 0.91 |   | 9.96 | 4.25 |    | 9.78 | 4.12 |
> |Moderate | CFR-A  |2.58 |0.85  |    | $\textbf{3.06}$ | $\textbf{1.02}$ | $ \uparrow$ | 9.89 | 4.22 |   | $\textbf{11.26}$ |$ \textbf{5.03}$| $ \uparrow$ |
> | shift | CFR-B  |$\textbf{2.98}$ |$ \textbf{0.99}$ | $ \uparrow$  | 2.65| 0.92 |   | $\textbf{12.35}$ | $\textbf{5.67} $| $ \uparrow$ | 9.83 | 4.18 |
>  | | CFR-C  | 2.56 | 0.85 |    | 2.63 | 0.90 |    | 9.88 | 4.21|   | 9.81 | 4.16 |
>  | | CERL  |2.59 | 0.86 |    | 2.66 | 0.92 |   | 9.90 | 4.24 |    | 9.82 | 4.17 |
> | No  | CFR-A | 2.58 | 0.87 |   | 2.62 | 0.88 |    | 9.86 | 4.20 |   | 9.85 | 4.19 |
> |shift  | CFR-B | 2.60 | 0.88 |   | 2.60 | 0.87 |   | 9.85 | 4.18 |   | 9.83 | 4.18 |
> | | CFR-C | 2.58 | 0.87 |    | 2.59 | 0.87 |    | 9.84 | 4.18|   | 9.83 | 4.18 |
> | | CERL | 2.59 | 0.87 |    | 2.60 | 0.87 |   | 9.85 | 4.19|   | 9.83 | 4.18 |
>
> The Table shows that, under no domain shift scenario, the three strategies and our model have a similar performance on the News and BlogCatalog datasets, because the previous and new data are from the same distribution. CFR-A, CFR-B, and CERL need fewer resources than CFR-C. Under substantial shift and moderate shift scenarios, we find strategy CFR-A performs well on previous data, but significantly declines on new dataset; strategy CFR-B shows the catastrophic forgetting problem where the performance on the previous dataset is poor; strategy CFR-C performs well on both previous and new data, but it re-trains the whole model using both the previous and new data. However, if there is a memory constraint or a barrier to accessing previous data, the strategy CFR-C cannot be conducted. Our CERL has a similar performance to strategy CFR-C, while CERL does not require access to previous data. Besides, by comparing the performance under substantial and moderate shift scenarios, the larger domain shift leads to worse performance of CFR-A and CFR-B. However, no matter what the domain shift is, the performance of our model CERL is consistent with the ideal strategy CFR-C.

---

> > ### Author Response · Authors · 2020-11-25
> > **Response to Reviewer 2 - Continued**
> >
> > **Review 3:**
> >
> > We very much appreciate the suggestion on the ablation study. We have conducted three ablation studies to evaluate the effectiveness of the important components in CERL, i.e., CERL (w/o FRT), CERL (w/o herding), and CERL (w/o cosine norm). CERL (w/o FRT) is the simplified CERL without the feature representation transformation, which is based on traditional continual learning with knowledge distillation and integral probability metrics. In CERL (w/o FRT), we do not store and transform the previous feature representation into new feature space, and only utilize the knowledge distillation to realize the continual learning task and balance the bias between treatment and control with each new data. CERL (w/o herding) adopts a random subsampling strategy to select samples into memory, instead of herding algorithm. CERL (w/o cosine norm) removes the cosine normalization in the last representation layer. From the Table, the performance becomes poor after removing anyone in the feature representation transformation, herding, or cosine normalization modules compared to the original CERL. More specifically, after removing the feature representation transformation, $\sqrt{\epsilon_\text{PEHE}}$ and $\epsilon_\text{ATE}$ increase dramatically, which demonstrates that the knowledge distillation always used in continual learning task is not enough for the continual causal effect estimation. In addition, using herding to select a representative set of samples from treatment and control distributions is crucial for the feature representation transformation.
> >
> >
> > ||Previous data|||New data|||
> > |: ----- :|: ----- :|: ----- :|: ----- :|: ----- :|: ----- :|: ----- :|
> > | Strategy     | $\sqrt{\epsilon_\text{PEHE}}$   | $\epsilon_\text{ATE}$ |   | $\sqrt{\epsilon_\text{PEHE}}$   | $\epsilon_\text{ATE}$ |
> > |    CFR-A | 1.47 | 0.35 |   |  $\textbf{2.51}$ |  $\textbf{0.73}$ | $ \uparrow$ |
> >  |   CFR-B |  $\textbf{1.82}$ |  $\textbf{0.47}$ | $ \uparrow$ | 1.63 | 0.45 |
> >  |   CFR-C | 1.49 | 0.36 |   | 1.62 | 0.44 |
> > |   CERL | 1.49 | 0.37 |    | 1.63 | 0.44 |
> >  |    CERL (w/o FRT) |  $\textbf{1.71}$ |  $\textbf{0.43}$ |$ \uparrow$  | 1.63 | 0.44|
> >   |   CERL (w/o herding) |  $\textbf{1.57}$ |  $\textbf{0.40}$ | $ \uparrow$ | 1.63| 0.44|
> >    |  CERL (w/o cosine norm) |  $\textbf{1.51}$ |  $\textbf{0.38}$ | $ \uparrow$ | 1.65 | 0.44|

---

### Official Review · AnonReviewer3 · 2020-10-28
**The work adopts continual lifelong learning for causal effect estimation. I have concerns on the motivation and the assumption. I also suggest several improvements for the experiments.**

**Rating:** 4
**Confidence:** 5

**Review:**

The work adopts a continual lifelong learning paradigm for causal effect estimation. In five synthetic datasets, the proposed method achieves similar results to CFR-C, a strong baseline which requires to store all data and needs to be trained from scratch.

I have some concerns about the motivation of this work. A simple question about the claim on the original data: why does the performance of the model trained continually on the original data matter? It is not difficult to save a checkpoint of the model trained on original data and use it for the original data whenever you want, given the fact the neural network based causal inference models are small. Or you can let the model do the inference and save the results periodically. Similarly, I could not get why “catastrophic forgetting” is an issue in this problem. If storing the original data would cause a copyright or privacy issue, I wonder why storing a model or feature representations trained by such data can avoid these issues. The memory issue of storing data is also not likely to happen in real-world since collecting data is often more complicated and expensive than storing them.

In section 2, there is an issue in the strong ignorability assumption. It would be great if the authors can clarify. The notation \mathcal{X}_d assumes that the feature space can change over domains, but the ignorability needs a unified feature space. So, the ignorability assumption actually relies on varying observed covariates space when $d$ changes.

Regarding Table 1, could the authors report the exact setting for the CERL model which achieves the reported numbers? For example, the number of stored feature representations. This is because in  Figure 4, it shows M=10000 has higher PEHE and \epsilon_ATE than the one reported in Table 1.

Since training from scratch with all data is expensive, it would be interesting to show a comparison of storage space, RAM memory and runtime between CERL and CFR-A, B and C.

---

> ### Author Response · Authors · 2020-11-25
> **Response to Reviewer 1**
>
> We are deeply grateful for your comments and propositions on how to improve our paper. We have addressed these points in the revised version.
>
> **Motivation**:
>
> The reviewer raised concerns on the motivation of our work, based on the assumption: "given the fact the neural network based causal inference models are small".
>
> We agree with the reviewer that the commonly used benchmarks (e.g., IHDP, Jobs, and Twins) for causal inference, as well as the trained models, are usually small. In particular, these benchmarks have the following properties: (1) all the data are available at a time and they cannot continue to grow; (2) all the data are from one single stationary source; (3) the size of data is usually small so that we don’t face memory constraint problem. Thus, it might be meaningless to apply continual causal effect estimation for these benchmark datasets.
>
> However, the newly proposed technique, continual causal effect estimation, is suitable for many real-world applications that continuously collect data from multiple sources. For instance, healthcare data are usually composed of multiple databases collected from different hospitals or other organizations even across multiple countries. And the data are incessantly growing to update the databases in real-time. In these practical scenarios, the motivation for applying continual causal effect estimation is two-fold. First, given the large size of data that is continuously collected, re-training models every time from scratch (when a new batch of data is collected) is impractical. Second, the group imbalance within each domain and the domain shifts over time present new research challenges for treatment effect estimation, which cannot be addressed by existing methods.
>
> **Assumptions**:
>
> Sorry for the confusion about the notation $\mathcal{X}_d$. We have deleted the notation $\mathcal{X}_d$. The observational data contain $n$ units collected from $d$ different domains and the $d$-th dataset $D_d$ contains the data $\\{(x,y,t) |x\in X, y\in Y, t\in T \\}$ collected from $d$-th domain. For $d$ datasets $\\{D_1, D_2,..., D_d\\}$, they have the common observed variables, i.e., they are in the same feature space. However, due to the fact that they are collected from different domains, they have different distributions with respect to $X$, $Y$, and $T$. For the whole data, the feature space is consistent. $\\{D_1, D_2,..., D_d \\}$ are only the subsets of the whole population. Therefore, the strong ignorability assumption still holds.

---

> > ### Author Response · Authors · 2020-11-25
> > **Response to Reviewer 1 - Continued**
> >
> > **Experiments**:
> >
> > We have added the new experiments based on real data and a series of ablation studies in the revised version of the manuscript. We also reported the exact setting for the experiments as suggested by the reviewer.
> >
> > As for the issue about limited experimental evidence, we have added two semi-synthetic benchmarks, i.e., News {johansson2016learning, schwab2018perfect} and BlogCatalog {guo2020learning} for continual causal effect estimation. For both datasets, we consider three scenarios to represent the different degrees of domain shifts among the incrementally available observational data, i.e., substantial shift, moderate shift, and no shift.
> >
> > |||   News      ||||||BlogCatalog       ||||||
> > |: - :|: - :|: - :|: - :|: - :|: - :|: - :|: - :|: - :|: - :|: - :|: - :|: - :|: - :|
> > |||   Previous data|||New data|||Previous  data|||New data|||
> > ||Strategy|  $\sqrt{\epsilon_\text{PEHE}}$|$\epsilon_\text{ATE}$| |$\sqrt{\epsilon_\text{PEHE}}$|$\epsilon_\text{ATE}$| |$\sqrt{\epsilon_\text{PEHE}}$|$\epsilon_\text{ATE}$| |$\sqrt{\epsilon_\text{PEHE}}$|$\epsilon_\text{ATE}$| |
> > |Substantial | CFR-A |2.49 | 0.80 |    | $\textbf{3.62}$ | $\textbf{1.18}$ | $ \uparrow$ | 9.92 | 4.25 |   | $\textbf{13.65}$ | $\textbf{6.21}$ |  $ \uparrow$|
> > |  shift | CFR-B  |$\textbf{3.23}$ | $\textbf{1.06}$| $ \uparrow$  | 2.71 | 0.91 |   |  $\textbf{14.21}$ | $\textbf{6.98}$ | $ \uparrow$ | 9.77 | 4.11 |
> > |     | CFR-C  | 2.51 | 0.82 |    | 2.70 | 0.92 |    |  9.93 | 4.24|   | 9.77 | 4.12 |
> > |   | CERL  |2.55 | 0.84 |    | 2.71 | 0.91 |   | 9.96 | 4.25 |    | 9.78 | 4.12 |
> > |Moderate | CFR-A  |2.58 |0.85  |    | $\textbf{3.06}$ | $\textbf{1.02}$ | $ \uparrow$ | 9.89 | 4.22 |   | $\textbf{11.26}$ |$ \textbf{5.03}$| $ \uparrow$ |
> > | shift | CFR-B  |$\textbf{2.98}$ |$ \textbf{0.99}$ | $ \uparrow$  | 2.65| 0.92 |   | $\textbf{12.35}$ | $\textbf{5.67} $| $ \uparrow$ | 9.83 | 4.18 |
> >  | | CFR-C  | 2.56 | 0.85 |    | 2.63 | 0.90 |    | 9.88 | 4.21|   | 9.81 | 4.16 |
> >  | | CERL  |2.59 | 0.86 |    | 2.66 | 0.92 |   | 9.90 | 4.24 |    | 9.82 | 4.17 |
> > | No  | CFR-A | 2.58 | 0.87 |   | 2.62 | 0.88 |    | 9.86 | 4.20 |   | 9.85 | 4.19 |
> > |shift  | CFR-B | 2.60 | 0.88 |   | 2.60 | 0.87 |   | 9.85 | 4.18 |   | 9.83 | 4.18 |
> > | | CFR-C | 2.58 | 0.87 |    | 2.59 | 0.87 |    | 9.84 | 4.18|   | 9.83 | 4.18 |
> > | | CERL | 2.59 | 0.87 |    | 2.60 | 0.87 |   | 9.85 | 4.19|   | 9.83 | 4.18 |
> >
> > Take the News dataset as one example. To generate two sequential datasets with different domain shifts, we combine the news items belonging to LDA topics from $1$ to $25$ into the first dataset and the news items belonging to LDA topics from $26$ to $50$ into the second dataset. There is no overlap of the LDA topics between the first and second datasets, which is considered as substantial shift. In addition, the news items belonging to LDA topics from $1$ to $35$ and items belonging to topics $16$ to $50$ are used to construct the first and second datasets, respectively, which create the moderate shift scenario. Finally, randomly sampled items from $50$ LDA topics compose the first and second dataset with no shift, because they are from the same distribution. Under each domain shift scenario and each dataset, we randomly sample $60\%$ and $20\%$ of the units as the training set and validation set and let the remaining be the test set.
> >
> >
> > The Table shows that, under no domain shift scenario, the three strategies and our model have a similar performance on the News and BlogCatalog datasets, because the previous and new data are from the same distribution. CFR-A, CFR-B, and CERL need fewer resources than CFR-C. Under substantial shift and moderate shift scenarios, we find strategy CFR-A performs well on previous data, but significantly declines on new dataset; strategy CFR-B shows the catastrophic forgetting problem where the performance on the previous dataset is poor; strategy CFR-C performs well on both previous and new data, but it re-trains the whole model using both the previous and new data. However, if there is a memory constraint or a barrier to accessing previous data, the strategy CFR-C cannot be conducted. Our CERL has a similar performance to strategy CFR-C, while CERL does not require access to previous data. Besides, by comparing the performance under substantial and moderate shift scenarios, the larger domain shift leads to worse performance of CFR-A and CFR-B. However, no matter what the domain shift is, the performance of our model CERL is consistent with the ideal strategy CFR-C.

---

> > > ### Comment · AnonReviewer3 · 2020-11-25
> > > **Thank the authors for their hard work on the response**
> > >
> > > Although some of my concerns are not fully covered by the response. I must say I appreciate the authors' very hard work on the response. It is impressive that the authors performed additional experiments on two larger semi-synthetic datasets.
> > >
> > > The authors' response resolves (1) part of my concern about motivation and (2) the ignorability assumption. The authors also tried to explain the cost of each model. It would be better if you can include solid results about the complexity/cost of each model.
> > >
> > > But as a practitioner, it might be hard for me to adopt a NN based causal effect estimation method with observational data if I really want to obtain a (likely) unbiased estimate from large-scale, continuous collected, across domain data, which is already expensive to collect and store. I would probably prefer to launch an online experiment or at least, use a solid model that has been widely adopted in the industry (e.g., synthetic control). This is just my personal opinion, which does not influence my score for this work.
> > >
> > > PS I would encourage the authors to submit the response a little earlier next time.

---

> > > > ### Author Response · Authors · 2020-11-25
> > > > **Thanks for your comment.**
> > > >
> > > > I really appreciate your prompt reply. Due to the rebuttal time limit, we cannot provide solid results about the complexity or cost for each experiment. We will make constant efforts to provide more details about experiments maybe in the later arXiv version. Thanks for your suggestions as a practitioner. In the industry,  the widely used and market-proven model should be a better choice for now. These NN based causal effect estimation method maybe can solve some new challenges in the future.

---

### Author Response · Authors · 2020-11-25
**Thanks to All Reviewers**

We thank the reviewers for taking the time to provide such detailed feedback and thoughtful suggestions. We address each of the reviewer’s questions and concerns individually. We have also uploaded an updated copy of the manuscript with the revisions we mention below.

---

### Decision · Program_Chairs · 2021-01-07
**Final Decision**

**Decision:**

Reject

**Comment:**

The authors consider the problem of causal inference from multiple conditionally ignorable models that yield different observed data distributions.  This problem is distinct from transportability (which assumes some types of causal invariance across domains, and aims to move causal conclusioned learned in one context to another using this invariance).  The authors adapt a machine learning approach from (Shalit et al, 2017).

Because the authors describe an algorithm rather than a model, it was a bit difficult to understand what assumptions tie the different observed data distributions together (I am guessing there is a way to formulate a 'global model' tying all datasets together in terms of the algorithm hyperparameters but the authors do not discuss this).

The authors evaluate their method via a simulation study.  Moreover, in response to reviewer criticism, the authors uploaded additional results from semi-synthetic data.

Some of the concerns of reviewers were about novelty and scope of evaluation (in addition, some complained about writing and notation).